# Multimodality Imaging of Moyamoya Disease: A Practical Guide for Neuroradiologists Based on a Case Report

**DOI:** 10.3390/reports8040232

**Published:** 2025-11-11

**Authors:** Elisa Ferraro, Agata Amaduri, Corrado Ini’, Mario Travali, Francesco Tiralongo, Pietro Valerio Foti, Concetto Cristaudo, Antonio Basile

**Affiliations:** 1Department of Medical Surgical Sciences and Advanced Technologies “G.F. Ingrassia”—Radiology I Unit, University Hospital Policlinico “G. Rodolico-San Marco”, Via Santa Sofia 78, 95123 Catania, Italy; elisaferraro86.ef@gmail.com (E.F.); agata.amaduri@gmail.com (A.A.); tiralongofrancesco91@hotmail.it (F.T.); pietrofoti@hotmail.com (P.V.F.); basile.antonello73@gmail.com (A.B.); 2UOC Neuroradiologia, Azienda Ospedaliera per l’Emergenza Cannizzaro, 95126 Catania, Italy; mario.travali.a34l@gmail.com (M.T.); concetto.cristaudo@gmail.com (C.C.); 3NANOMED-Research Centre for Nanomedicine and Pharmaceutical Nanotechnology, University of Catania, 95125 Catania, Italy; 4Centro di Ricerca Multidisciplinare “Chirurgia delle Sindromi Malformative Complesse della Transizione e dell’Età Adulta” (ChiSMaCoTA), Department of Medical Surgical Sciences and Advanced Technologies “G.F. Ingrassia”, University of Catania, 95123 Catania, Italy

**Keywords:** moyamoya disease, neuroradiology, magnetic resonance, computed tomography, angiography

## Abstract

**Background and Clinical Significance**: Moyamoya disease is a rare, progressive cerebrovascular disease characterized by steno-occlusion of the terminal internal carotid arteries and the arteries around the circle of Willis, with the formation of abnormal collateral vessels. Early clinical manifestations include recurrent hemodynamic transient ischemic attacks (TIAs), especially in young subjects. Multimodality imaging, including computed tomography, magnetic resonance, and digital subtraction angiography, is necessary to reach a correct diagnosis in young patients with stroke-like symptoms. Various radiological findings are crucial for early diagnosis, staging, and management of moyamoya disease. **Case Presentation**: We describe the case of a 31-year-old male presenting with acute focal neurological deficits and a history of recurrent TIAs. Neuroimaging was performed to assess vascular pathology, parenchymal injury, and collateral circulation and to provide critical information on vascular anatomy and the extent of ischemic damage. **Conclusions**: The purpose of this case report is to illustrate the main specific radiological signs and the diagnostic value of multimodality neuroimaging in the evaluation of moyamoya disease, providing a practical imaging-based diagnostic approach for neuroradiologists.

## 1. Introduction and Clinical Significance

First reported by Takeuchi and Shimizu in 1957 and then fully described by Suzuki and Takaku in 1969, moyamoya disease (MMD) is a progressive vaso-occlusive disease characterized by stenosis and/or occlusion of the terminal portion of the internal carotid artery (ICA) and the arteries around the circle of Willis, with the development of an abnormal vascular network [1,2]. Although the disease is also known as “spontaneous occlusion of the circle of Willis”, the International Classification of Diseases recognizes “moyamoya” as the specific name for this condition because the characteristic appearance of the network of collateral vessels on imaging has been compared to “something hazy, like a puff of cigarette smoke”, which, in Japanese, is moyamoya [3].

The incidence of MMD is higher in East Asian countries (Japan, Korea, and China) than in Europe, America, and Africa, and, according to a survey conducted in Japan in 2004, the prevalence of MMD was approximately 6.03/100,000 with an estimated incidence of 0.35/100,000 [2,3]. The male to female ratio is 1:2.2, and there are two peaks of incidence, at 10–20 and 35–50 years of age [2,4]. Familial cases account for approximately 10–15% of patients, and the risk for family members of subjects with MMD is substantially higher than the general population [5]. Clinical presentation of MMD is insidious, and varies depending on the age and the area of the brain involved [2]. According to the “Classification of the Japanese Health Ministry”, there are four clinical forms of moyamoya disease: ischemic, hemorrhagic, epileptic, and “other” [6]. In pediatric and young adult patients, the primary clinical manifestations are recurrent hemodynamic transient ischemic attacks (TIAs) or ischemic stroke, due to cerebral hypoperfusion resulting from a progressive occlusion of major intracranial vessels. Conversely, the hemorrhagic form is more prevalent in adults [2,6]. Early diagnosis and accurate evaluation of MMD are crucial to improve the prognosis of patients and significantly reduce the risk of further cerebrovascular events and long-term disability [7,8]. Since stroke-like symptoms are unusual in young people, imaging plays a pivotal role in the early diagnosis and management of MMD, enabling the characterization of both the primary pathology and associated complications, and ruling out other potential differential diagnoses. Different neuroimaging techniques, including computed tomography angiography (CTA), magnetic resonance imaging (MRI), magnetic resonance angiography (MRA), digital subtraction angiography (DSA), single-photon emission CT (SPECT), and positron emission tomography (PET), have revolutionized the diagnostic process of MMD by providing detailed insights into vascular anatomy, collateral circulation, and cerebral perfusion dynamics [7]. Furthermore, the integration of advanced MRI techniques, such as perfusion-weighted imaging (PWI), has enhanced our ability to assess hemodynamic compromise, which is essential for surgical planning and prognostication [9]. According to the latest guidelines released by the “Research Committee on Spontaneous Occlusion of the Circle of Willis (Moyamoya Disease)” in 2021, MMD can be diagnosed with cerebral angiography or magnetic resonance angiography (MRA) [8].

We describe the case of a 31-year-old young male with the onset of sudden focal neurological symptoms and with radiological findings suggesting MMD. We also reviewed more specific features on CT, MR, and DSA, focusing on main radiological signs, and providing practical guidance for neuroradiologists approaching this condition. The case we report conforms to the Consensus-based Clinical Case Reporting Guidelines (CARE) [10].

## 2. Case Presentation

A 31-year-old Chinese male was referred to the emergency department of our hospital with acute onset of motor deficit in the left upper limb, worsening of flexion stance, headache, and increasing language impairment; these symptoms developed approximately 6 h before admission and progressed over the following 3 h to reach a stable moderate neurological deficit. His medical history revealed a previous and recurrent ischemic stroke, multiple TIAs over the previous two years, characterized by recurrent episodes of unilateral limb weakness and transient speech disturbances, all of which resolved spontaneously within a few hours. No previous neuroimaging had been performed, and no family history of cerebrovascular disease, seizures, developmental impairment, or congenital anomalies was reported. There was also no history of smoking, alcohol consumption, or prior head and neck irradiation.

Neurological examination revealed a left-sided weakness (Medical Research Council grade 3/5), mild expressive aphasia, and a National Institutes of Health Stroke Scale (NIHSS) score of 8 points, suggestive of moderate stroke. Routine laboratory tests, including coagulation parameters and inflammatory markers, were within normal limits. Hemoglobin, red blood cells, white blood cells, platelets, hematocrit, glucose, magnesium, calcium, lipase, bilirubin, alkaline phosphatase, urea, ammonia, sodium, potassium, chloride, prothrombin time, and homocysteine were all within normal values. Blood and urine cultures were all normal. C-reactive protein was elevated at 11.23 mg/L (normal range 0.0–5 mg/L), fibrinogen was elevated at 639 mg/L (normal range 170–400 mg/L), and activated partial thromboplastin time (aPTT) was elevated at 41.1 s (normal range 24–36 s). Vital parameters were within normal limits (blood pressure: 125/90 mm Hg, heart rate: 80/min, temperature: 36.8 °C, respiratory rate: 18/min, oxygen saturation: 97% on room air). ASA therapy and antiedema measures were started on admission after the CT scan; however, intravenous thrombolytic therapy was not performed because the onset of symptoms was beyond the therapeutic window.

The patient underwent multi-detector computed tomography (MDCT, Optima 64 slice, GE Healthcare, Milwaukee, WI, USA) (Appendix A) of the brain for suspected stroke. A CT scan revealed the presence of cortico-subcortical hypodensities in the right frontal and parietal regions, in the deep white matter of the right periventricular area, associated with swelling of the cerebral cortex and obliteration of subarachnoid spaces (Figure 1).

These findings suggested acute ischemic lesions in the right middle cerebral artery territory. There were no hemorrhagic lesions or mass effect on the encephalic parenchyma. Computed tomography angiography (CTA) examination showed reduced caliber and parietal irregularity of M1–M2 tracts of the middle cerebral artery bilaterally, with steno-occlusive tract-like changes. Additionally, vascular ectasia of deep lenticular arteries and the right anterior cerebral artery were also detected (Figure 2).

CT findings raised the suspicion of vaso-occlusive disease, and, in particular, moyamoya disease; to exclude secondary causes of moyamoya disease, the following parameters were also evaluated: autoimmune panel (ANA, ENA, anti-dsDNA, and ANCA), thyroid function and antibodies (TSH, FT3, FT4, and anti-TPO), infectious screening (HBV, HCV, HIV, and syphilis), and thrombophilia panel; all of these parameters showed normal values. Subsequently, the patient underwent MRI, MRA, and DSA examinations. MR examination (performed with a Signa HDxT MR scanner, GE Healthcare, Milwaukee, WI, USA) showed the presence of hyperintense areas on T2-weighted and FLAIR sequences, with true restricted diffusion (increased signal on DWI sequences and reduced ADC values), located in right prefrontal, superior frontal, and postcentral gyrus; in the right periventricular and supraventricular white matter; and at the level of the splenium of the corpus callosum (Figure 1); the cerebral cortex also showed a swollen appearance in these regions, confirming recent ischemic lesions. A small hypointense area on T2*-weighted sequences was also identified at the level of the aforementioned altered signal area of the right supraventricular white matter. This area was also hyperintense on T1-weighted sequences, indicating a subacute hemorrhagic lesion. Moreover, chronic ischemic lesions, characterized by hyperintensity on T2-weighted and FLAIR sequences, with no diffusion restriction, were detected in the right parietal, occipital, and temporal cortico-subcortical regions (Figure 3).

Prominent leptomeningeal collaterals with slow flow, resulting in high signal intensity on FLAIR sequences, were detected in the subarachnoid spaces of the left hemisphere (“ivy sign”) (Figure 4).

MRA confirmed the reduced caliber and wall irregularities with steno-occlusive changes in M1–M2 tracts of the middle cerebral artery bilaterally, predominantly on the right. The A2 segment of the right anterior cerebral artery (ACA) and the P2 segment of the right posterior cerebral artery (PCA) also showed irregular caliber due to alternations of stenotic and dilated tracts (Figure 2).

To confirm the diagnosis and to assess disease staging, DSA was performed using a six-vessel approach (ICAs, ECAs, and vertebral arteries), revealing significant stenosis of both M1 segments of the middle cerebral arteries, more prominent on the right, associated with mild progressive caliber reduction in supraclinoid internal carotid arteries (“champagne bottle neck” sign) and abnormal moyamoya vessels within basal ganglia and thalami, represented by hypertrophic lenticulostriate and thalamoperforating arteries (“puff of smoke” sign) (Figure 5).

Multiple and focal stenoses were detected at the origins of M2 branches and the proximal A1 segment of the right ACA, also with the involvement of P1–P2 segments of both PCAs. The blood supply to the cerebral hemispheres was allowed by collateral circulation through leptomeningeal anastomoses from posterior circulation and anterior cerebral arteries, and through choroidal and perforator arteries arising from M1 and P1. Collateral supply through the external carotid arteries was absent. According to DSA findings and excluding secondary causes of MMS, the patient was classified as Suzuki stage III of idiopathic MMD.

Unfortunately, additional techniques such as SPECT/PET were not performed, as they were not available at our institution.

## 3. Discussion

The term moyamoya, meaning “smoke-like” in Japanese, is used to describe the appearance of cerebral vessels on imaging. MMD predominantly involves anterior circulation and is characterized by spontaneous progressive bilateral occlusion of the terminal tracts of internal carotid arteries (ICAs) [9]. The affected vessels do not exhibit atherosclerotic or inflammatory alterations; instead, they demonstrate an overgrowth of the smooth muscle layer with fibrocellular proliferation and thickening of the intima. The progressive occlusion of the arteries leads to the development of a prominent network of compensation collaterals, arising from intracranial internal carotid arteries, posterior cerebral arteries, or anterior choroidal arteries and reaching distal branches of post-stenotic middle cerebral arteries. Collaterals appear as wide and winding vessels along the course of the thalamo-striatal arteries and lenticulo-striatal arteries. Thromboses of these vessels lead to ischemic lesions, while microaneurysms increase the risk of intracranial hemorrhage, a not uncommon finding in MMD [6]. The association of other conditions (Down’s syndrome, neurofibromatosis, autoimmune diseases, brain tumors, meningitis, head irradiation), which may cause similar cerebrovascular lesions, should be distinguished from MMD and should be referred to as “moyamoya-like” or “moyamoya syndrome” [11,12].

### 3.1. Pathophysiology and Differential Diagnosis

The pathogenic mechanisms of MMD are not fully understood; nevertheless, endothelial and smooth muscle cell proliferation, with narrowing of the arterial outer diameter and fibrocellular thickening of the intima, leads to vascular occlusion and aberrant angiogenesis, resulting in the formation of moyamoya vessels. Genetic factors, such as modifications in the Ring Finger 213 (RNF213) gene, involved in mediating protein–protein interactions, or alterations in microRNAs, may play a pivotal role, along with environmental influences. Figure 6 illustrates the potential mechanisms of moyamoya disease.

The differential diagnosis of idiopathic (“true”) moyamoya disease includes other conditions developing occlusive vasculopathies and leading to imaging appearances similar to MMD. In such cases, the underlying condition is referred to as moyamoya syndrome (MMS) rather than moyamoya disease [3]. Several medical conditions are associated with moyamoya syndrome, such as Down syndrome, neurofibromatosis type 1 (NF1), radiotherapy to the head or neck (particularly radiotherapy for optic gliomas, craniopharyngiomas, and pituitary tumors), sickle cell anemia, and thyroid disease. Rarer conditions encompassing MMS also include connective tissue disorders and autoimmune diseases (systemic lupus erythematosus, antiphospholipid syndrome), infections (tuberculosis and bacterial meningitis, viral vasculitis), and genetic conditions (Turner syndrome, Noonan syndrome, and Alagille syndrome) [13,14]. Even if idiopathic moyamoya disease presents a specific imaging pattern, since it typically affects both supraclinoid ICAs and spares the posterior circulation, epidemiologic data, laboratory tests, and imaging features are mandatory in young subjects with stroke-like symptoms to rule out other possible differential diagnoses. Therefore, it is imperative that physicians have a comprehensive understanding of MMS and the ability to make a differential diagnosis, as this facilitates the management of the disease, the determination of the most suitable treatment, and a more profound comprehension of the pathogenesis of MMD [13].

### 3.2. Clinical Aspects and Radiological Features

Clinically, moyamoya symptoms are primarily associated with the brain arterial vascular territories involved, manifesting as hemiparesis, cognitive impairment, aphasia, visual deficits, and seizures. Headache is a common symptom in MMD, possibly due to the dilatation of meningeal collaterals. These symptoms can be triggered by stress events, such as hyperventilation or dehydration. Hemorrhage is also a frequent complication of the disease in adults, while choreiform movements have mainly been observed in children [3].

Radiological imaging plays a crucial role in the diagnosis and evaluation of moyamoya disease, mainly in young patients presenting with stroke-like symptoms. In recent years, advancements in non-invasive medical imaging have led to the development of more sophisticated methods for evaluating the characteristic vascular alterations of MMD. Currently, cross-sectional imaging (CT, CTA, MR, MRA) is used to assess these alterations, offering a more comprehensive approach to the diagnosis [6]. According to the “Revised Diagnostic Criteria for Moyamoya disease”, the diagnosis of unilateral cases must be made through angiography, while the diagnosis of bilateral cases can already be obtained using angiography or magnetic resonance. Therefore, MRI and DSA are part of the diagnostic criteria [11].

An emergent CT examination is the first imaging modality to distinguish ischemic stroke from intracranial hemorrhage. In MMD, ischemic lesions may be observed in the basal ganglia, in the deep white matter, subcortically, and in the periventricular regions, commonly involving the watershed zone. Acute ischemic lesions manifest with parenchymal hypodensity, blurring and indistinctness of the gray-white matter junction, decreased density of the basal ganglia, and cortical sulcal effacement. Nevertheless, patients with only transient ischemic attacks may exhibit a normal CT scan result. The second step is determining whether a major cerebral vessel is stenotic or occluded. CTA is the procedure of choice to identify stenoses and occlusions of large intracranial vessels, as well as collateral vasculature from deep arteries or pial collateral. The presence of these abnormalities, particularly in young people, may be indicative of moyamoya disease. Therefore, in the setting of an emergency or where MRI is not readily available, CTA should be the first diagnostic option [3,6].

Magnetic resonance imaging is a valuable diagnostic tool even for patients with early-stage disease, offering the capability to accurately determine diagnosis, playing a crucial role in evaluating dimensions and phase of ischemic lesions with a specificity and sensitivity superior to CT [2,6]. Conventional MR examination includes T1-, T2-weighted, FLAIR, DWI, SWI, and TOF sequences (Appendix A), and it can show direct and indirect signs of MMD. MRA and CTA detect stenosis and occlusion mainly affecting distal segments of ICAs, with narrowing of both supraclinoid ICAs, and proximal segments of middle cerebral arteries, as well as the presence of prominent collateral vessels [7]. In MMD, the middle cerebral artery (MCA) exhibits a reduced outer diameter and diminished wall thickness, accompanied by a conspicuous presence of collateral vessels, differentiating MMD from other vasculopathies. Moreover, the appearance of shrinkage of the middle cerebral artery may serve as an indicator of disease progression, since it has been proven that the outer diameter of the MCA appears smaller in different stages, particularly the early and late phases of the disease [15]. High-resolution vessel wall imaging are different imaging techniques that require high spatial resolution or suppressing signal from flowing blood, and they can assess disease activity by evaluating not only wall thickening but also diminished wall enhancement [16,17]. One of the most compelling characteristics of the disease on MR is the reduction in flow voids within the vessels involved, accompanied by pronounced vascular flow voids in deep vascular territories, mainly the thalamus and basal ganglia [9].

The “ivy” sign is an indicator of slow or retrograde flow in engorged pial vessels, via leptomeningeal collaterals, and is characterized by linear hyperintensity on FLAIR sequences in cerebral sulci of the affected brain region, as in the case we report. The ivy sign is also detected on contrast-enhanced T1-weighted sequences, and it may be less pronounced on 3D FLAIR compared to 2D FLAIR sequences, probably because of the differential impact of flow velocity in the 2D versus 3D FLAIR imaging; it has also been hypothesized that the uniform suppression of the cerebrospinal fluid signal inherent to 3D FLAIR may obscure both the abnormal thickening and/or signal intensity of the leptomeninges, as well as pathological signal alterations within the CSF itself [18,19]. In the case we report, this sign was more evident in the brain hemisphere opposite to the ischemic lesion. This apparent paradox could be explained by the rapid onset of arterial occlusion in the right middle cerebral artery that did not allow the development of compensatory collaterals [20]. The ivy sign has been associated with cerebrovascular reactivity and decreased vascular reserve, and, following revascularization surgery, it can improve or worsen temporarily due to hyperperfusion [14]. Moreover, an elevated Suzuki stage (≥3), CBF reduction, and high risk of severe ischemic events, correlate independently with the ivy sign [21].

Another phenomenon that can be observed on FLAIR sequences is the “medullary streak” sign, which is characterized by a hyperintense streak oriented perpendicularly to the lateral ventricle (“periventricular medullary veins/FLAIR hyperintense streaks”); despite its etiology remaining ambiguous, the condition has been associated with ischemia and could represent collateral vasculature, stagnated cerebrospinal fluid, or vasogenic edema (Figure 7).

Approximately half of adult patients with MMD may develop intracranial hemorrhage. The location of the hemorrhage can be subarachnoid, intraventricular, or intraparenchymal, more often within the basal ganglia [3]. Microbleeds can be detected through the use of gradient echo T2* sequences or susceptibility-weighted imaging (SWI), the latter being a 3D gradient-echo sequence with an increased spatial resolution and higher sensitivity to these alterations. Microbleeds may be associated with leakage from dilated and fragile collateral vessels, such as the anterior choroidal or posterior communicating arteries. These alterations also have a prognostic value, as they indicate an increased risk for intraventricular hemorrhage, particularly if located in the periventricular area [2]. In fact, rupture of prominent and fragile anterior choroidal arteries leads to intraventricular hemorrhage, as often observed in MMD [22]. Furthermore, SWI and FFE sequences can detect prominent medullary veins, identifying hypointense blooming artifacts within multiple intracranial blood vessels (“prominent vessel sign”). This sign reflects an increased concentration of paramagnetic deoxyhemoglobin in cortical veins (referred to as the “cortical vein sign”), subependymal and medullary veins (referred to as the “brush sign”) (Figure 8), and it is associated with an increased risk of infarction, low cerebral blood flow (CBF), and low cerebrovascular reactivity [17,23].

Beyond conventional sequences, MR also uses sophisticated techniques to discern alterations in microstructure and connectivity. This technique incorporates diffusion tensor imaging (DTI), showing lowered FA and elevated ADC in the pathologic regions of the brain, and diffusion kurtosis imaging; however, the latter requires longer scanning times and is not as diffuse as DTI [17].

In summary, MRI diagnostic criteria for moyamoya disease necessitate the following: occlusion or stenosis of the terminal portion of the ICA; reduction in the outer diameter of the terminal portion of the ICA and the horizontal portion of MCAs in T2 weighted sequences; and abnormal vessels in the basal ganglia and/or periventricular area. Deep abnormal vessels can be detected when two or more flow voids are visualized in the periventricular white matter or in the basal ganglia, even if unilateral [11].

MRI and CT can identify indirect signs of MMD, such as ill-defined encephalomalacic/CSF density areas (previous ischemic lesions), cortical atrophy, gliosis, and white matter deterioration [4,16]. The assessment of cerebral hemodynamics can be facilitated by the implementation of perfusion imaging techniques. CT perfusion is a procedure available on a global scale; its application requires the administration of high doses of radiation and the use of iodinated contrast agent. Dynamic susceptibility contrast (DSC- MRI) is the most frequently used MR perfusion; it is considered beneficial for the analysis of potential surgical outcomes in MMD. The prolonged mean transit time (MTT) in DSC-MRI may signal the presence of misery perfusion, which is a compensatory increase in the oxygen extraction fraction (OEF) that occurs when the regulatory system is unable to attain an adequate cerebral blood flow (CBF) due to a decrease in cerebral perfusion pressure [7]. Arterial spin labeling (ASL-MRI) uses endogenous water as a natural tracer, and this technique has been demonstrated to evaluate post-surgical hemodynamic dysfunction in MMD [9]. Patients with MMD will generally exhibit decreased CBF, increased cerebral blood volume (CBV) and OEF, augmented MTT, and reduced cerebral vascular reactivity [16]. Positron emission tomography (PET) is a useful technique for evaluating CBF, CBV, OEF, and cerebral metabolic rate of oxygen (CMRO). It provides insights into metabolic demands, offering prognostic information. It has been proposed that adults and pediatric patients affected by MMD and with hemispheric lesions may demonstrate a decrease in CMRO, yet exhibit a substantial enhancement of CMRO following revascularization [7]. Single photo-emission CT (SPECT) has been used to evaluate CBF and cerebral vascular reserve before and after treatment. SPECT evaluation indicated that patients with MMD are more prone to post-revascularization hyperperfusion compared to those with other treated cerebrovascular diseases [7].

Bilateral and, especially, unilateral MMD need confirmation with angiography, also allowing treatment planning. DSA should be performed using a five-vessel or six-vessel approach, including imaging of both external and internal carotid arteries, and one or both vertebral arteries [3,9]. In the context of preoperative imaging, the assessment of the external carotid arteries is paramount for the identification of preexisting collateral vessels, especially at the level of the orbit through the ophthalmic artery and the posterior and anterior ethmoidal arteries; this is of critical importance to avoid their disruption during surgical intervention [24]. Angiography identifies narrowing of proximal and extracranial segments of ICA (“bottle neck” sign), and abnormal proliferation of lenticulostriate and thalamoperforator vessels, giving the characteristic “puff of smoke” appearance (Figure 5). Furthermore, aneurysms and any arteriovenous malformations associated with moyamoya are most effectively detected through conventional angiography rather than CT or MR [3]. In 2021, the revised version of the “Diagnostic Criteria for Moyamoya Disease” delineated the following diagnostic angiographic findings of MMD [11]:– Stenosis or occlusion involving the terminal segment of the ICA.– The presence of moyamoya vessels, in proximity to the stenotic artery during the arterial phase.

Suzuki et al. developed a system for organizing the progression of MMD into six phases, with the basis of this system being the dynamic changes in the affected vessels [1,9]. There is an observable range of vessel functionality, ranging from mild ICA narrowing to complete disappearance of the moyamoya collaterals and occlusion of the ICA, which consequently leads to exclusive reliance on external carotid artery circulation as follows:Stage I: narrowing of the internal carotid artery bifurcation.Stage II: presence of visible moyamoya collateral vessels, dilated ACAs, MCAs, and narrowed ICA bifurcation.Stage III: augmentation of moyamoya vessels and narrowed ACA and MCA.Stage IV: diminished number of moyamoya vessels, deterioration of ICA, ACA, MCA, and presence of collateral from external carotid artery.Stage V: occlusion of ICA, ACA and MCA; further deterioration of moyamoya vessels; and increased collateral from extracranial vessels.Stage VI: loss of moyamoya collaterals, ACA, and MCA. Persistence of collaterals from the external carotid arteries or vertebral arteries [7,9].

DSA also has the potential to facilitate the prediction of clinical outcomes and prognosis in MMD. Choroidal anastomosis and posterior cerebral artery involvement may act as risk factors for hemorrhagic lesion development. Furthermore, the longitudinal shift in collateral blood vessels from the anterior to the posterior intracranial circulation may suggest the onset of hemorrhagic stroke events. The current literature has demonstrated the unique role of angiography in evaluating MMD, determining its hemodynamic aspects, and assessing the presence of a compensation network [16].

Main radiological signs of MMD with their frequency of detection on MR, CTA, and DSA are summarized in Table 1.

### 3.3. Management and Outcome

Treatment for moyamoya disease encompasses both medical and surgical therapies. The main purpose of therapy is to improve blood flow to the brain and prevent further stroke events. Oral antiplatelet therapy has been conventionally used worldwide among asymptomatic patients with MMD. Even if some studies reported a reduced risk of hemorrhagic stroke, the evidence for antiplatelet treatment is low, and this therapy did not demonstrate a reduced risk of ischemic stroke. The reason for this assumption is that ischemic insult in patients with MMD is not embolic, secondary to endothelial damage and platelet adhesion, but due to hemodynamic alterations. However, medical therapies are used as supportive and symptom control treatments, and they may include acetylsalicylic acid, calcium channel blockers, and anti-seizure medications. Surgical revascularization is the mainstay of treatment for MMD to prevent cerebral stroke and restore reserve capacity of the brain, improving CBF. Surgical options include direct revascularization, indirect revascularization, or combined approaches. Direct revascularization creates a bypass of the ischemic regions, connecting a branch of the external carotid artery (the superficial temporal artery) to a superficial branch of the MCA. This technique provides an immediate restoration of blood flow. Indirect revascularization uses connective or muscle tissues placed on the surface of the brain to stimulate the formation of new blood vessels. Indirect surgical techniques include encephalomyosynangiosis (EMS), encephaloduroarteriosynangiosis (EDAS), encephalomyoarteriosynangiosis (EMAS), encephaloduroarteriomyosynangiosis (EDAMS), and encephalogaleosynangiosis (EGS). These approaches are relatively easier to perform than direct ones, but need more time to improve cerebral blood flow [25]. Early surgical revascularization may reduce the risk of rebleeding in patients presenting with intracranial hemorrhage, and resulted in better outcome than nonoperative management. On the other hand, in patients with ischemia, different studies did not provide conclusive results, with some suggesting delaying revascularization to reduce mortality rate, and others suggesting revascularization within a few months from diagnosis, especially in children [26].

Overall, the prognosis of patients with MMD depends on multiple factors. Patients diagnosed in childhood, or adults with different comorbidities (diabetes mellitus, smoker), and those initially presenting with ischemic stroke have a worse prognosis and an increased risk of recurrent symptoms; such patients could benefit better from early revascularization treatments [27].

## 4. Conclusions

Moyamoya disease is often characterized by non-specific clinical symptoms, particularly in early stages, and a prompt diagnosis in young patients with stroke-like symptoms is a crucial issue in the therapeutic pathway of cerebrovascular disorders. Imaging is essential for timely recognition and accurate diagnosis of the disease, enabling the identification of specific radiological signs, including characteristic steno-occlusive changes to the terminal internal carotid arteries and the development of compensatory vascular collateral networks. Imaging techniques are also useful for disease staging, treatment planning, and follow-up. Raising awareness of imaging hallmarks and specific radiologic signs of moyamoya disease among radiologists and clinicians is fundamental to improving early diagnosis and timely intervention, preventing irreversible neurological damage, and improving patient outcomes.

## Figures and Tables

**Figure 1 reports-08-00232-f001:**
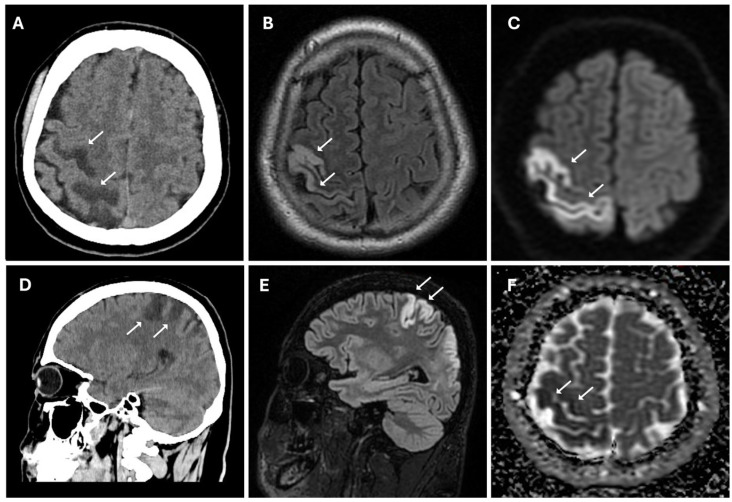
Acute ischemic lesion in the right frontoparietal region involving both the precentral and postcentral gyri. (**A**,**D**) Non-enhanced CT scan in axial (**A**) and sagittal (**D**) plans show hypodensity area (white arrows) in the right frontoparietal cortex and subcortical white matter, consistent with early ischemic changes. (**B**,**E**) Axial (**B**) and sagittal (**E**) 3D FLAIR sequences show hyperintense signal (white arrows) in cortical and subcortical areas, indicative of edema in the context of an evolving infarct. (**C**,**F**) Diffusion-weighted imaging (**C**) reveals marked hyperintensity (white arrows), while the corresponding apparent diffusion coefficient map (**F**) shows hypointensity (white arrows), confirming true restricted diffusion compatible with acute ischemia (cytotoxic edema).

**Figure 2 reports-08-00232-f002:**
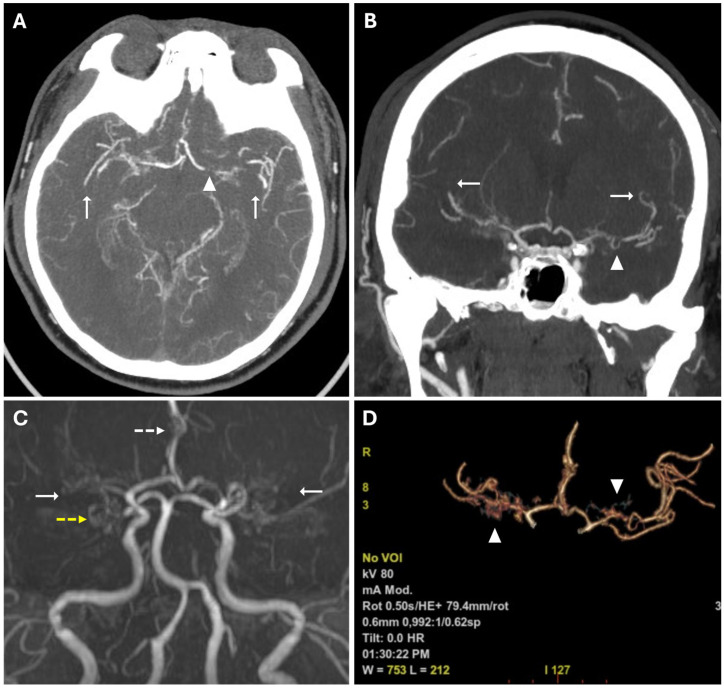
(**A**,**B**) CTA images in axial (**A**) and coronal (**B**) views, using a Maximum Intensity Projection (MIP) reconstruction, show reduced caliber and parietal irregularity of the M1-M2 tract bilaterally (white arrows) with severe stenotic tract in the left MCA (white arrowheads). (**C**) 3d MRA sequence shows multiple steno-occlusive changes in M1-M2 bilaterally (white arrows), irregular caliber of the right ACA (white dotted arrows), and right PCA (yellow dotted arrows). (**D**) Volume rendering reconstruction of CTA shows abnormal collateral perforators (white arrowheads).

**Figure 3 reports-08-00232-f003:**
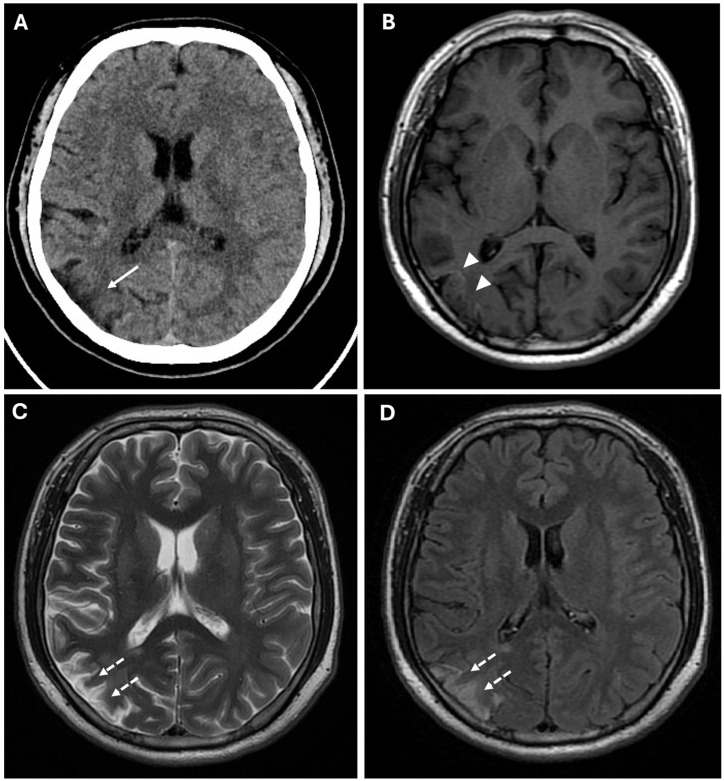
Chronic ischemic lesions. (**A**) A non-enhanced CT image in axial view shows a hypodensity area in the right frontal and parietal lobes with atrophic brain cortex (white arrow). (**B**) Axial T1-weighted spin echo sequence shows mild linear hyperintensity in the right parietal lobe, coherent with cortical laminar necrosis (white arrowheads). (**C**,**D**) Axial T2-weighted (**C**) and FLAIR (**D**) sequences show right parietal and periventricular areas of hyperintensity compatible with subacute-chronic ischemic lesions (white dotted arrows).

**Figure 4 reports-08-00232-f004:**
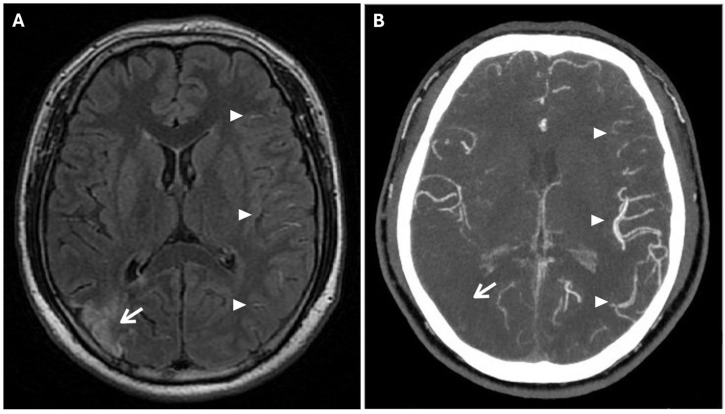
The Ivy sign in the left-brain hemisphere. (**A**) Axial 2D FLAIR sequence shows linear hyperintensity along the cortical sulci of the left hemisphere (white arrowheads); note a subacute ischemic lesion in the right parietal lobe (white arrow). (**B**) CTA Maximal Intensity Projection (MIP) shows prominent leptomeningeal collaterals in the left hemisphere, with abnormal enhancement of blood vessels, “ivy-like” (white arrowheads); the absence of opacification of distal branches of the right MCA (white arrow) was consistent with recent ischemic stroke in the right parietal lobe vascular territory.

**Figure 5 reports-08-00232-f005:**
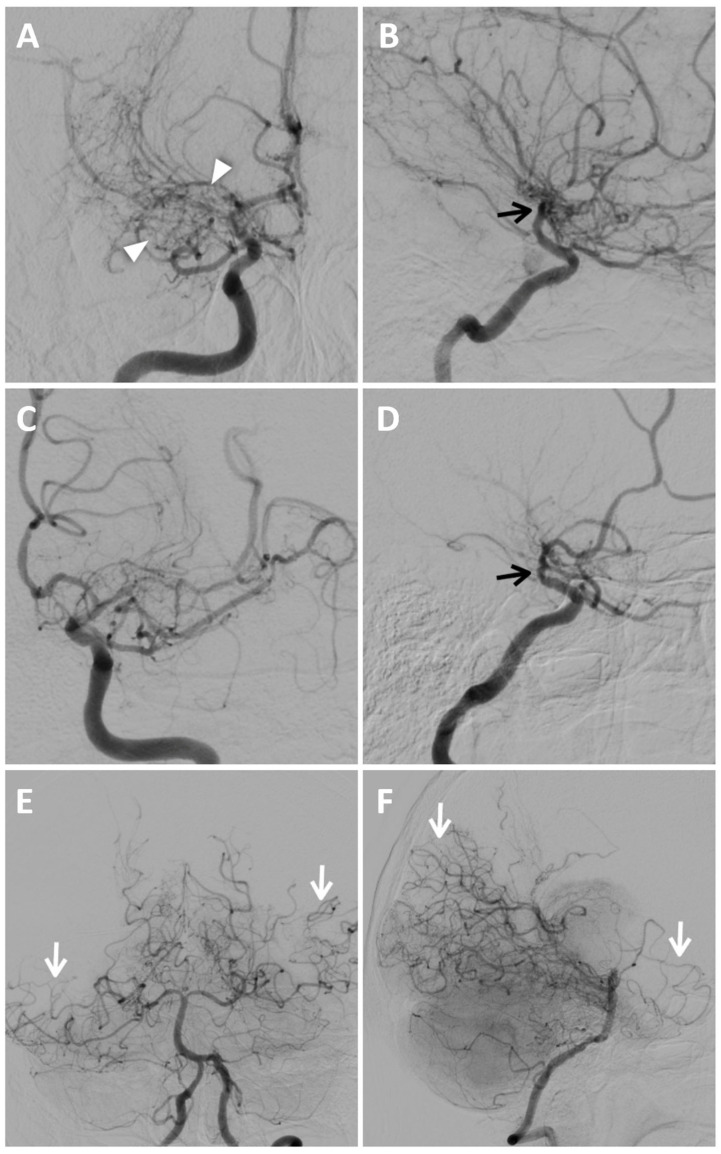
Digital subtraction angiography of MMD. Anteroposterior (**A**) and lateral views (**B**) of the right internal carotid artery injection show extensive and severe stenosis of the M1 tract of MCA and progressive caliber reduction in supraclinoid segment of the ICAs (“champagne bottle neck” sign, black arrows); notice the wide perforator network represented by lenticulostriate and thalamic arteries, resembling a “puff of smoke” (white arrowheads). Anteroposterior (**C**) and lateral views (**D**) of the left internal carotid artery injection demonstrate mild stenosis of the supraclinoid segment of the ICA (black arrow in (**D**)), of the distal portion of the M1 segment of MCA and multiple focal stenoses involving the ACA and M2; moyamoya vessels are less evident than on the right side. Anteroposterior (**E**) and lateral views (**F**) of posterior circulation injection show collateral pathways through posterior lenticulostriate arteries at the basal ganglia level, and leptomeningeal collaterals are more prominent in the right temporal lobe and left temporoparietal region (white arrows).

**Figure 6 reports-08-00232-f006:**
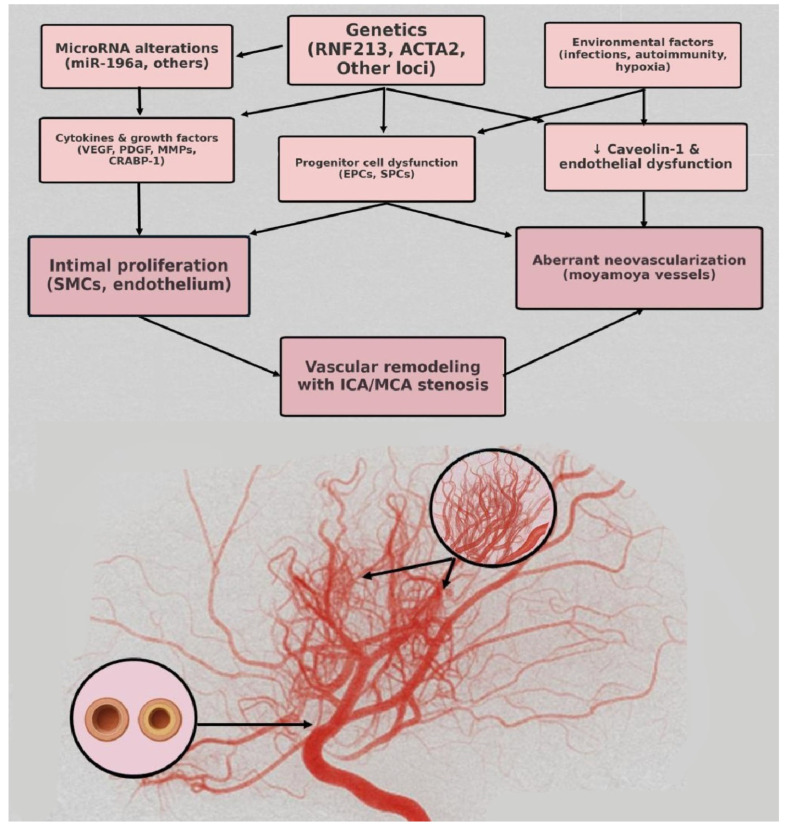
Pathophysiology mechanisms of moyamoya disease.

**Figure 7 reports-08-00232-f007:**
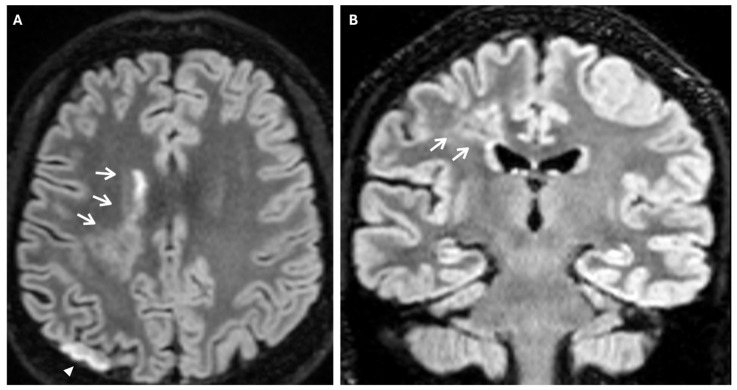
Axial (**A**) and coronal (**B**) 3d FLAIR sequences show a hyperintense streak area oriented perpendicularly to the right lateral ventricle (white arrows), representing the “medullary streak” sign. Note the right parietal subacute ischemic lesion (white arrowhead).

**Figure 8 reports-08-00232-f008:**
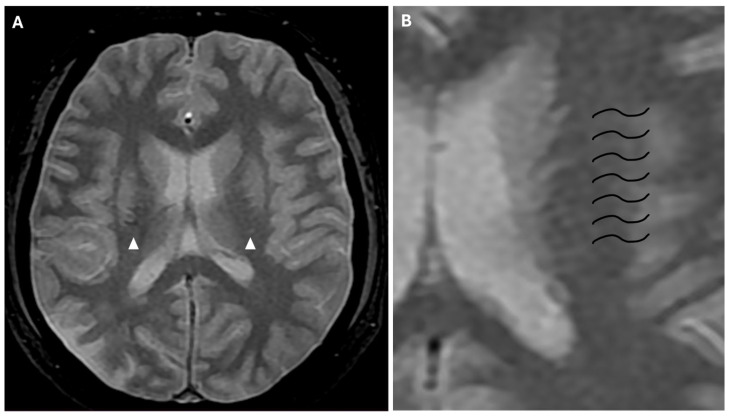
(**A**) Axial FFE sequence shows subependymal and medullary veins bilaterally, representing the “prominent vessel sign” (white arrowheads). (**B**) Magnification of the previous image illustrates deep medullary veins on the left hemisphere (wavy black lines).

**Table 1 reports-08-00232-t001:** Radiological findings and detection rate of MMD through imaging.

Radiological Findings	CT/CTA	MRI	DSA
Bottle neck sign	+ + +	+ +	+ + +
Ivy sign	+	+ + +	+
Puff of smoke sign	+	+	+ + +
Medullary streak sign	+	+ + +	+ +
Prominent vessel sign	+	+ + +	+
Cortical vein sign	+	+ + +	+ +
Brush sign	+	+ + +	+ +

+: low detection rate; + +: medium detection rate; + + +: high detection rate.

## Data Availability

The original contributions presented in this study are included in the article. Further inquiries can be directed to the corresponding author.

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
