# Peer review of "Multimodality Imaging of Moyamoya Disease: A Practical Guide for Neuroradiologists Based on a Case Report"

_reports, 2025, doi:10.3390/reports8040232_

Round 1

Reviewer 1 Report

Comments and Suggestions for Authors

Thanks for the invitation to review this work. This is an interesting report. I have several comments for authors to consider:

  1. There is a mismatch in the abstract and the main manuscript. The abstract claims imaging “guid[ed] both medical and surgical therapeutic decision-making,” yet no actual treatment choices, peri-procedural plans, or outcomes are reported later (medical therapy, revascularization type/timing, or follow-up). Add a short “Management & Outcome” paragraph (acute phase, secondary prevention, revascularization decision [direct STA-MCA vs indirect EDAS/encephaloduroarteriosynangiosis] and early outcomes).
  2. Some details vis-à-vis acute care management are missing. On arrival: NIHSS 8 and “acute onset,” but there’s no onset-to-door/time-to-imaging, IVT/EVT eligibility rationale, BP/oxygenation/temperature targets, antithrombotics, or antiedema measures. Please add a brief acute stroke timeline and decision logic (e.g., in the setting of bilateral M1 stenosis, why IVT/EVT was/was not pursued).
  3. Authors state that CARE CARE checklist was adhered. However, the checklist is incomplete; for example, the case lacks a timeline, differential work-up, and follow-up. Please include details on a minimal timeline, a succinct differential (ICAD, vasculitis, dissection, radiation/autoimmune/thyroid-associated MMS), and disposition/follow-up.
  4. “Routine labs normal,” but no targeted work-up for moyamoya syndrome mimics (autoimmune, thyroid, prior irradiation, infectious, hematologic; optional RNF213) is described. Add a sentence on negative secondary-cause evaluation to support “idiopathic MMD.” 
  5. Authors stage the case as Suzuki III with right-predominant MCA disease and also report a left-hemisphere ivy sign. That can be physiologically plausible (contralateral hypoperfusion) but needs explanation, especially since stenoses are worse on the right. Add a one-line clarification that Ivy sign reflected leptomeningeal slow flow in the less stenotic hemisphere due to steal or asymmetric CVR (see Ravindran et al, Eur J Neurosci. 2021). 
  6. Author's note “collateral supply through the external carotid arteries was absent,” yet in Suzuki III, ECA collaterals may be emerging. Clarify whether ECA injections were performed and explicitly state five- or six-vessel DSA protocol adherence. (Authors correctly recommend 5–6 vessel DSA elsewhere; make the patient’s acquisition explicit.)
  7. Authors discuss DSC/ASL/SPECT/PET, but didn’t perform any. Given surgical planning implications, add (if available) DSC or ASL with acetazolamide CVR, or state that perfusion wasn’t feasible acutely and is planned pre-revascularisation.
  8. In Figure 4, label clearly that edema/infarct is right parietal while ivy sign is left; explain the paradox in legend to pre-empt confusion.
  9. “Medullary streak sign” is an uncommon parlance; authors do cite it. Consider aligning with “periventricular medullary veins/FLAIR hyperintense streaks” and clearly differentiate FLAIR streaks (Figure 6) from SWI “brush sign” (Figure 7).
  10. Some more discussion is needed vis-à-vis acute phase management. Please state antithrombotic choice in light of a small subacute hemorrhagic focus on T2*/T1 (how this influenced aspirin/dual therapy/anticoagulation deferral). 
  11. The manuscript needs significant editing for spelling errors and overall flow. 

Author Response

We wish to take opportunity to thank the Reviewers for their comments regarding our paper, and for the changes they suggested, that could improve the quality of the manuscript. A response to reviewers’ comments was underlined in bold. In the manuscript, modifications suggested by the Reviewer 1 were highlighted in yellow, and those suggested by the Reviewer 2 in green. We hope the revised version of the manuscript may now be suitable for publication in Reports journal.

Kind regards,

The authors

REVIEWER 1

Thanks for the invitation to review this work. This is an interesting report. I have several comments for authors to consider:

  1. There is a mismatch in the abstract and the main manuscript. The abstract claims imaging “guid[ed] both medical and surgical therapeutic decision-making,” yet no actual treatment choices, peri-procedural plans, or outcomes are reported later (medical therapy, revascularization type/timing, or follow-up). Add a short “Management & Outcome” paragraph (acute phase, secondary prevention, revascularization decision [direct STA-MCA vs indirect EDAS/encephaloduroarteriosynangiosis] and early outcomes).

I agree with the comments of reviewer 1; a “Management & Outcome” paragraph has been added to the discussion section, describing the main type of treatments available and their outcomes. However, the aim of our paper is to describe the main diagnostic features and radiological findings of MMD; detailed information about treatment goes beyond the purpose of this case report, and therefore were not fully described.

  1. Some details vis-à-vis acute care management are missing. On arrival: NIHSS 8 and “acute onset,” but there’s no onset-to-door/time-to-imaging, IVT/EVT eligibility rationale, BP/oxygenation/temperature targets, antithrombotics, or antiedema measures. Please add a brief acute stroke timeline and decision logic (e.g., in the setting of bilateral M1 stenosis, why IVT/EVT was/was not pursued).

We are grateful to the reviewer for this thoughtful and constructive comment. In the revised version, we have added a brief acute stroke timeline describing the symptom onset, time of hospital arrival, and the initial management upon admission, including vital sign monitoring and supportive measures. We have also specified that antiplatelet therapy with acetylsalicylic acid (ASA) was initiated on admission, while intravenous thrombolysis was not performed because the patient presented more than six hours after symptom onset, thus outside the therapeutic window.
These details have been included in the revised “Case Presentation” section of the manuscript.

  1. Authors state that CARE CARE checklist was adhered. However, the checklist is incomplete; for example, the case lacks a timeline, differential work-up, and follow-up. Please include details on a minimal timeline, a succinct differential (ICAD, vasculitis, dissection, radiation/autoimmune/thyroid-associated MMS), and disposition/follow-up.

The CARE checklist has been completed and updated.

  1. “Routine labs normal,” but no targeted work-up for moyamoya syndrome mimics (autoimmune, thyroid, prior irradiation, infectious, hematologic; optional RNF213) is described. Add a sentence on negative secondary-cause evaluation to support “idiopathic MMD.” 

In the “case presentation” section, routine labs analyzed have been reported, and differential diagnosis were considered in the targeted work-up for MMD mimics.

  1. Authors stage the case as Suzuki III with right-predominant MCA disease and also report a left-hemisphere ivy sign. That can be physiologically plausible (contralateral hypoperfusion) but needs explanation, especially since stenoses are worse on the right. Add a one-line clarification that Ivy sign reflected leptomeningeal slow flow in the less stenotic hemisphere due to steal or asymmetric CVR (see Ravindran et al, Eur J Neurosci. 2021).

In our case, the ivy sign was mainly observed in the left hemisphere, indicating significant leptomeningeal activation. This finding was absent in the right hemisphere, which may have contributed to brain ischemia due to hypoperfusion.

  1. Author's note “collateral supply through the external carotid arteries was absent,” yet in Suzuki III, ECA collaterals may be emerging. Clarify whether ECA injections were performed and explicitly state five- or six-vessel DSA protocol adherence. (Authors correctly recommend 5–6 vessel DSA elsewhere; make the patient’s acquisition explicit.)

DSA protocol was performed using a six-vessel approach (ICAs, ECAs and vertebral arteries).

  1. Authors discuss DSC/ASL/SPECT/PET, but didn’t perform any. Given surgical planning implications, add (if available) DSC or ASL with acetazolamide CVR, or state that perfusion wasn’t feasible acutely and is planned pre-revascularisation.

We thank the reviewer for this valuable comment. As suggested, we clarified in the revised version that ASL and DSC-MRI were not feasible in the acute setting since it was not available at our institution at that moment.

  1. In Figure 4, label clearly that edema/infarct is right parietal while ivy sign is left; explain the paradox in legend to pre-empt confusion.

The apparent paradox has been explained in the discussion section.

  1. “Medullary streak sign” is an uncommon parlance; authors do cite it. Consider aligning with “periventricular medullary veins/FLAIR hyperintense streaks” and clearly differentiate FLAIR streaks (Figure 6) from SWI “brush sign” (Figure 7).

Medullary streak sign is detected on FLAIR sequences as hyperintense streak in the periventricular white matter. This sign should be associated with collateral circulation, stagnated cerebrospinal fluid, vasogenic edema or a combination of these conditions (Suzuki H, Mikami T, Kuribara T, Yoshifuji K, Komatsu K, Akiyama Y, Ohnishi H, Houkin K, Mikuni N. Pathophysiological consideration of medullary streaks on FLAIR imaging in pediatric moyamoya disease. J Neurosurg Pediatr. 2017 May;19(5):560-566. doi: 10.3171/2017.1.PEDS16541. Epub 2017 Mar 10. PMID: 28291429). On the other hand, the brush sign reflects an increased concentration of paramagnetic deoxyhemoglobin in draining veins and capillaries manifesting as hypointensity within multiple intracranial blood vessels, as described in the discussion section.

  1. Some more discussion is needed vis-à-vis acute phase management. Please state antithrombotic choice in light of a small subacute hemorrhagic focus on T2*/T1 (how this influenced aspirin/dual therapy/anticoagulation deferral). 

Antithrombotic therapy was suspended after MRI. 

  1. The manuscript needs significant editing for spelling errors and overall flow. 

The text has been extensively reviewed by a native English speaker, and we hope it is now clear and accessible to all readers who wish to make use of it.

Reviewer 2 Report

Comments and Suggestions for Authors

Thank you for inviting me to review this submission titled “Multimodality imaging of moyamoya disease: a practical guide for neuroradiologists based on a case report”. Here are some comments and suggestions for the authors:

  1. This is an interesting manuscript showing a rational approach stemming from the radiological POV for Moyamoya disease. Although it is based on a report, I found it very useful for readers.
  2. The introduction is clear. The terms and updates regarding Moyamoya disease (MMD) are well explained.
  3. The case is well-written and adequately described.
  4. The images are well illustrated and explained. The main concept of each study is clear. I’d suggest adding a separate subsection if you want to describe the acquisition parameters of each study.
  5. Revise typing errors all over the manuscript, like in the figure 5 legend "sovraclinoid”.
  6. The manuscript may benefit from a graphical illustration explaining the pathophysiology of the disease.
  7. The discussion is pertinent, and the main concepts among the different study types are mentioned and discussed.
Comments on the Quality of English Language

Revise typing errors.

Author Response

We wish to take opportunity to thank the Reviewers for their comments regarding our paper, and for the changes they suggested, that could improve the quality of the manuscript. A response to reviewers’ comments was underlined in bold. In the manuscript, modifications suggested by the Reviewer 1 were highlighted in yellow, and those suggested by the Reviewer 2 in green. We hope the revised version of the manuscript may now be suitable for publication in Reports journal.

Kind regards,

The authors

REVIEWER 2

Thank you for inviting me to review this submission titled “Multimodality imaging of moyamoya disease: a practical guide for neuroradiologists based on a case report”. Here are some comments and suggestions for the authors:

  1. This is an interesting manuscript showing a rational approach stemming from the radiological POV for Moyamoya disease. Although it is based on a report, I found it very useful for readers.
  2. The introduction is clear. The terms and updates regarding Moyamoya disease (MMD) are well explained.
  3. The case is well-written and adequately described.
  4. The images are well illustrated and explained. The main concept of each study is clear. I’d suggest adding a separate subsection if you want to describe the acquisition parameters of each study.

CT and MRI protocols were included as supplementary files.

  1. Revise typing errors all over the manuscript, like in the figure 5 legend "sovraclinoid”.

All typing errors were revised. The text has also been extensively reviewed by a native English speaker.

  1. The manuscript may benefit from a graphical illustration explaining the pathophysiology of the disease.

We thank the reviewer for this helpful suggestion. Following the recommendation, we added a new graphical illustration summarizing the main pathophysiological mechanisms of moyamoya disease. The figure (now included in the “discussion” section) integrates genetic, molecular, and environmental factors leading to endothelial dysfunction, intimal proliferation, and aberrant neovascularization, ultimately resulting in vascular remodeling and ICA stenosis. This visual aid aims to provide a clearer overview of the disease process and to enhance the educational value of the manuscript.

The discussion is pertinent, and the main concepts among the different study types are mentioned and discussed.

Round 2

Reviewer 2 Report

Comments and Suggestions for Authors

Thank you for adressing my queries. This new version is remakable better and sufficient for publication.